# Effects of Fatty-Type and Lean-Type on Growth Performance and Lipid Droplet Metabolism in Pekin Ducks

**DOI:** 10.3390/ani12172268

**Published:** 2022-09-01

**Authors:** Zhong Zhuang, Tingshuo Yang, Wenqian Jia, Meng Bai, Hao Bai, Zhixiu Wang, Guohong Chen, Yong Jiang, Guobin Chang

**Affiliations:** 1Key Laboratory for Animal Genetics & Molecular Breeding of Jiangsu Province, College of Animal Science and Technology, Yangzhou University, Yangzhou 225009, China; 2Joint International Research Laboratory of Agriculture and Agri-Product Safety, Ministry of Education of China, Institutes of Agricultural Science and Technology Development, Yangzhou University, Yangzhou 225009, China

**Keywords:** Pekin duck, fatty-type, lean-type, lipid deposition, lipid droplet

## Abstract

**Simple Summary:**

Lipid deposition in animals is closely related to lipid anabolism. In order to further explore how differences in different metabolic types regulate lipid deposition, we compared the growth performance and lipid droplet metabolism of fatty-type ducks and lean-type ducks. The results showed that fatty-type ducks showed a faster growth rate and more fat deposition in the early growth stage after feeding the same diet, and produced more lipoproteins in serum and deposited in adipose tissue. However, fewer triglycerides accumulated in the liver. We believe that this performance of fatty-type ducks is caused by the increased expression level of lipid droplet-related genes.

**Abstract:**

The reasons for differences in lipid depositions between fatty-type (F-T) and lean-type (L-T) ducks remain unknown. The present study aimed to compare the growth performance, lipid deposition, and gene expression related to lipid droplet formation in F-T and L-T Pekin ducks. One-day-old, 140 each L-T and F-T male ducks were selected and distributed separately into 20 replicate cages. All ducks were fed commercial diets up to 35 d of age. F-T ducks had a higher average daily gain from 21 to 28 d of age. On 35-day-old, F-T ducks had higher serum levels of high- and low-density lipoprotein cholesterol, cholesterol, albumin, and hydroxybutyrate dehydrogenase activity than L-T ducks. F-T ducks had higher abdominal fat and subcutaneous fat percentages than those in L-T ducks. Liver histological examination showed that L-T ducks contained more lipid droplets in the liver, which gradually decreased with increasing age. The average adipocyte area and diameter of abdominal fat and subcutaneous fat in the F-T and L-T ducks increased with age and were higher in F-T ducks than those in L-T ducks. Furthermore, the gene expression of perilipin 1, perilipin 2, angiopoietin-like protein 4, adipose triglyceride lipase, alpha/beta-hydrolase domain-containing protein 5 (ABHD5), and serine/threonine kinase 17a in the liver, abdominal fat, and subcutaneous fat of F-T ducks was higher than that in L-T ducks, and it increased with age. Compared to L-T ducks, F-T ducks had higher expression of ABHD5 in the abdominal fat and subcutaneous fat and lower expression in the liver. Thus, F-T ducks displayed lower hepatic lipid deposition and a higher percentage of abdominal fat and subcutaneous fat, suggesting that F-T ducks had higher lipid storage capacity due to increased gene expression related to lipid droplets.

## 1. Introduction

The Pekin duck, a high-quality local meat duck, shows excellent performance, rapid growth, and strong adaptability. Recently, Pekin ducks have become the primary meat duck breed for the production and improvement of duck meat in various countries globally [1]. To meet various consumer needs for different raw food constituents, the Chinese Academy of Agricultural Sciences bred two Pekin duck strains with different performance characteristics, referred to as lean-type (**L-T**) and fatty-type (**F-T**) ducks, which vary in lipid deposition and meat yield. Past research has shown that L-T ducks had 2.52% higher pectoral muscles and 0.46% higher leg muscles than F-T ducks in 35 d [2]. This could be attributed to enhanced meat deposition in L-T ducks, which improves protein anabolism and reduces protein catabolism [3]. The fat content in F-T ducks is 6–20% higher than that in L-T [2,3,4,5]. However, what causes the difference in lipid deposition between F-T and L-T ducks has not yet been determined.

In animals, the liver and adipose tissue are the main sites of lipid synthesis deposition [6,7]. The liver of poultry is primarily involved in fat metabolism and regulates lipid generation through precise and complex signaling [8]. Triglycerides synthesized in the liver are transported to muscle or adipose tissue by binding to apolipoproteins. It has been shown that 95% of fat in chicks is present in adipose tissue and other organs [9]. In addition, triglycerides (**TG**) are deposited in the adipose tissue in the form of lipid droplets, which play a key role in the balance between lipogenesis and lipolysis. Recently, it has been found that the excessive accumulation of lipid droplets leads to individual obesity, while insufficient accumulation of lipid droplets in adipocytes leads to serious changes in systemic energy metabolism, such as fat malnutrition [10]. The surface of lipid droplets is covered by coating proteins such as perilipin [11] and alpha/beta-hydrolase domain-containing protein 5 (**ABHD5**) [12], which stabilize the lipid droplets. Therefore, we speculated that the differential expression of lipid droplet synthesis-related genes in liver and adipose tissue of F-T ducks and L-T ducks may be one of the reasons affecting their lipid synthesis and metabolic differences.

The aim of this study was to determine the growth performance, fat deposition, serum biochemical parameters, liver, and adipose histology, and gene expression related to lipid droplet metabolism in F-T and L-T Pekin ducks, and to investigate the role of gene expression related to lipid droplets in regulating lipid deposition in Pekin ducks.

## 2. Materials and Methods

### 2.1. Experiments and Animal Handing

The ducks and care used in this experiment were approved by the Chinese Animal Care and Use Committee, and all procedures were performed under guidelines developed and approved by the Animal Care and Use Committee of Yangzhou University (approval number: 151-2014), throughout which we minimized animal pain, tissue damage, and stress.

One-day-old 180 Pekin male ducklings, 140 F-T ducks, and 140 L-T ducks were housed in 20 cages composed of wire floor enclosures (200 × 100 × 40 cm), 14 ducks per cage, and All birds were maintained under constant light and kept at 30 °C for 1 to 3 d, followed by a gradual decrease to 20 °C to 21 d. All cages had nipple drinkers and tubular feeders and were fed commercial diets ad libitum, until the end of 5 weeks. 

There were two commercial diets in this experiment, divided into edible diets for ducklings aged 1 to 21 days and 22 to 35 days, purchased from Hope Feed Co., Ltd. (Fangshan District, Beijing, China), and the main raw materials and nutritional components guaranteed by the products are shown in Table 1. Both F-T and L-T ducks were fed the same commercial diet.

### 2.2. Sample Collection

At 21, 28, and 35 d, the body weight and feed intake of the ducks in each cage were recorded after fasting for 12 h with ad libitum access to water. At the initial age of the 14 one-day-old ducks per cage (20 cages, 10 cages each for F-T and L-T), two ducks per cage were selected for slaughter at 21, 28, and 35 d based on average body weight, and immediately sacrificed via cervical dislocation after carbon dioxide anesthesia. Liver, abdominal fat, and subcutaneous fat were collected from a portion of Pekin ducks and fixed in 4% tissue fixative followed by Oil Red O or HE staining. The liver, abdominal fat, and subcutaneous fat of the remaining Pekin ducks were immediately stored in liquid nitrogen at −80 °C, until subsequent RT-PCR analysis. In addition, one Pekin duck per cage was selected at 35 days based on average body weight for calculation of slaughter performance using abdominal fat and subcutaneous fat collected. At 35 d, blood samples were collected by wing vein puncture with heparinized syringes equipped with stainless steel needles. The blood samples were centrifuged at 1000× *g* for 15 min at 4 °C to separate the upper transparent and stored at −20 °C until the measurement of serum biochemical parameters.

### 2.3. Measurement of Serum Biochemical Parameters

The concentrations of glucose, Triacylglycerol (**TG**), high-density lipoprotein cholesterol (**HDL-C**), low-density lipoprotein cholesterol (**LDL-C**), cholesterol (**CHOL**), uric acid, total bilirubin, direct bilirubin, total protein, albumin (**ALB**), and lactate dehydrogenase, were measured using commercial kits (Maccura Biotechnology Co., Ltd., Chengdu, China). The enzymatic activities of creatine kinase, hydroxybutyrate dehydrogenase, alkaline phosphatase, alanine aminotransferase, and aspartate aminotransferase were measured using an automatic analyzer (7080, Hitachi, Tokyo, Japan).

### 2.4. Histological Studies

The liver, abdominal fat, and subcutaneous fat samples were examined histologically using hematoxylin and eosin (**HE**) staining. These samples were washed in PBS, soaked in 4% paraformaldehyde for 24 h, removed, dehydrated, embedded in paraffin wax, and cut into slices of approximately 5 µm. These slices were collected on polylysine-coated glass slides and stained with HE or Oil Red O. The samples were scanned using a Nanozoomer scanner and the adipocyte area was calculated using an image analysis system (3DHISTECH’s Slide Converter).

### 2.5. RNA Extraction and Real-Time Quantitative PCR

Total RNA from the liver, abdominal fat, and subcutaneous fat was extracted using the TRIzol reagent (TaKaRa, Shanghai, China). One microgram of RNA from each sample was reverse-transcribed into cDNA using a reverse transcription kit (TaKaRa, Shanghai, China). Quantitative real time-PCR was performed in 96-well plates using an ABI 7500 real-time PCR system (Applied Biosystems, Foster City, CA, USA). Each well contained 2 µL of cDNA template (diluted 20× after the PCR reaction), 5 µL SYBR green (Bio-Rad, Foster City, CA, USA), and 1 µL each of the forward and reverse gene-specific primers at 10 µmol/L (primers are listed in Table 2). Glyceraldehyde 3-phosphate dehydrogenase was used as an endogenous reference gene for normalization. The thermal profile was 95 °C for 30 s followed by 40 amplification cycles at 95 °C for 15 s and 60 °C for 30 s. Subsequently, the melting curve analysis was performed. The resulting cycle threshold values were used to determine relative gene expression.

### 2.6. Statistical Analyses

All data, including growth performance, slaughter performance, serum biochemical indices, adipose tissue characteristics, and gene expression levels by RT-PCR, are expressed as mean ± SE. SPSS software (version 18.0; SPSS, Inc., Chicago, IL, USA) was used to compare the significance of repeated measurements between different growth ages and strains. *p* < 0.05 was used as the statistical significance criterion. Duncan’s multiple range test was used to detect and analyze the main effects among different growth ages and different strains.

## 3. Results

### 3.1. Growth Performace

In terms of body weight, L-T ducks at 21, 28, and 35 days were lower (*p* < 0.01, Figure 1a). The average daily feed intake of F-T ducks was higher than those of L-T ducks throughout the experimental period (*p* < 0.01; Figure 1b). There was no difference in average daily gain between F-T and L-T ducks at 28 to 35 d of age (*p* = 0.146); however, F-T ducks had a higher average daily gain from 21 to 28 d of age than that in L-T ducks (*p* < 0.05) (Figure 1c). By calculation, the feed to gain ratio of F-T ducks is higher than L-T ducks throughout the growth period (*p* < 0.01) (Figure 1d).

### 3.2. Serum Biochemical Parameters

F-T ducks exhibited higher serum concentrations of HDL-C, LDL-C, CHOL, ALB, and hydroxybutyrate dehydrogenase than L-T ducks did (*p* < 0.05, Table 3). Serum total bilirubin levels and creatine kinase activity were higher in the L-T ducks than in F-T ducks (*p* < 0.05, Table 3). The serum levels of glucose, TG, uric acid, direct bilirubin, and total protein and the activities of lactate dehydrogenase, alkaline phosphatase, alanine aminotransferase, and aspartate aminotransferase between F-T and L-T ducks have no differences (*p* > 0.05).

### 3.3. Liver Histology

Compared with the liver parenchyma cells of the F-T ducks, those of the L-T ducks showed widespread deposition of lipid droplets on Oil Red O staining at 21, 28, and 35 d (Figure 2). As age progressed, the number of hepatic lipid droplets decreased in both the F-T and L-T ducks.

### 3.4. Fat Deposition and Adiopocyte Histology

Compared with L-T ducks, F-T ducks had higher percentages of abdominal fat and subcutaneous fat at 35 d (*p* < 0.01, Figure 3), and HE staining showed that the adipocyte sizes of abdominal fat and subcutaneous fat in F-T ducks were higher (Figure 4).

Adipocyte diameter, area, and density of abdominal fat and subcutaneous fat were affected by age and strain in both F-T and L-T ducks (Table 4). These parameters were also affected by the interaction between age and strain in subcutaneous fat but not in abdominal fat (Table 4). As age progressed, adipocyte diameter and area increased, and adipocyte density decreased in abdominal fat in both F-T and L-T ducks (Table 4). F-T ducks had higher adipocyte diameter and area and lower adipocyte density in the abdominal fat than those in L-T ducks (Table 4). The adipocyte diameter and area in the subcutaneous fat of F-T ducks were higher than those in L-T ducks at 28 and 35 d (*p* < 0.05), whereas the adipocyte density was lower than that in L-T ducks at 28 and 35 d.

### 3.5. Gene Expression

In the liver, age and strain affected the expression of perilipin 1 (***PLIN1***), perilipin 2 (***PLIN2***), angiopoietin-like protein 4 (***ANGPTL4***), adipose triglyceride lipase (***ATGL***), *ABHD5*, and serine/threonine kinase 17a (***STK17A***). The interaction between age and strain affected the expression of *PLIN2* and *ATGL*, but had no influence on the expression of *PLIN1*, *ANGPTL4*, *STK17A*, and *ABHD5*. As age progressed, the mRNA levels of *PLIN1*, *PLIN2*, *ANGPTL4*, *ATGL*, and *STK17A* markedly increased (*p* < 0.01), but *ABHD5* gradually decreased (*p* < 0.01, Figure 5). The expression levels of *PLIN1*, *PLIN2*, *ANGPTL4*, *ATGL,* and *STK17A* were higher in F-T ducks than those in L-T ducks. *ABHD5* mRNA levels were higher in L-T ducks than those in F-T ducks (*p* < 0.01; Figure 5). 

In abdominal fat, age and strain also affected the expression of *PLIN1*, *PLIN2*, *ANGPTL4*, *ATGL*, *STK17A*, and *ABHD5*. The interaction between age and strain affected *PLIN1*, *ANGPTL4*, and *ABHD5* and had no influence on the expression of *PLIN2*, *ATGL*, and *STK17A*. The expression of *PLIN1*, *PLIN2*, *ANGPTL4*, *ATGL*, *ABHD5*, and *STK17A* in the abdominal fat was markedly upregulated throughout development (*p* < 0.01, Figure 6). The expression of *PLIN1*, *PLIN2*, *ANGPTL4*, *ATGL*, *STK17A*, and *ABHD5* in abdominal fat and F-T ducks was always higher than that in L-T ducks (*p* < 0.05, Figure 6).

In subcutaneous fat, age and strain affected the expression of *PLIN1*, *PLIN2*, *ANGPTL4*, *ATGL*, *STK17A*, and *ABHD5*. However, the interaction between age and strain affected *PLIN1*, *PLIN2*, *ATGL*, and *ABHD5* but had no influence on the gene expression of *ANGPTL4* and *STK17A*. As age progressed, the mRNA levels of *PLIN1*, *PLIN2*, *ANGPTL4*, *ATGL*, and *ABHD5* were markedly upregulated (*p* < 0.01); however, *STK17A* showed no major expression trend (*p* > 0.05, Figure 7). The expression levels of *PLIN1*, *PLIN2*, *ANGPTL4*, *ATGL*, *ABHD5*, and *STK17A* were higher in F-T ducks than those in L-T ducks (*p* < 0.05, Figure 7).

## 4. Discussion

In the present study, it was observed that F-T ducks had higher proportions of abdominal fat and subcutaneous fat than those in L-T ducks, which is similar to the findings of previous studies [2,3,5]. It has been observed that the F-T broilers accumulate more lipids than do the L-T broilers [13] and pigs [14]. Although we found a higher feed intake in F-T ducks, this might not be the main reason for the high percentage of abdominal fat and subcutaneous fat in these ducks. Reduced feed intake does not influence hepatic lipid deposition [2]. Therefore, genetic selection for meat yield reduces lipid deposition, which may be the main reason [15]. 

Lipid deposition is characterized by the enlargement of adipocytes, which results from increased TG storage. The major sources of TG deposition in adipose tissue are chylomicrons and very low-density lipoproteins. In the present study, L-T ducks showed higher amounts of lipid droplets in the liver than those in the F-T ducks. Interestingly, Saadoun and Leclercq has previously reported similar results [16]; prior to 5 weeks of age, the liver fat content of lean line chickens was marginally higher than that of fat line chickens. F-T ducks strongly express proteins related to glycolysis, ATP synthesis, and protein catabolism, whereas proteins highly expressed in L-T ducks reduce protein catabolism [3]. This indicates that F-T and L-T ducks have different liver metabolic capacities. In addition, the serum concentrations of HDL-C, LDL-C, CHOL, and ALB were higher in F-T ducks than those in L-T ducks. Obese animals usually have higher serum TG, LDL-C, and HDL-C levels [17,18,19,20]. LDL is the main form of liver lipid transport to extrahepatic tissues. Therefore, we speculate that the lipids synthesized by F-T Pekin duck livers are more rapidly transported to adipose tissue for storage than those synthesized by the L-T duck livers, resulting in less lipid deposition in the F-T duck liver.

As lipid droplets accumulate in adipocytes, adipocytes undergo hypertrophy (increase in size) and hyperplasia (increase in number) [21], which in turn produces expansion of adipose tissue [22]. In this study, as age progressed, the adipocyte size of the abdominal fat and subcutaneous fat increased, which were higher in F-T ducks than those in L-T ducks. Accumulating evidence suggests that adipocyte hyperplasia in broiler-type chicken occurs from 3 d post-hatching to 7 weeks of age, and hypertrophic growth occurs before 5 weeks of age [23], This is the same as our results. In addition, compared to lean animals, in both poultry and mammals, adipose tissue development of fatty type animals increased in both adipocyte formation and cell enlargement [24,25,26], suggesting that selecting for the fatness trait results in cellularity and volume changes in adipose tissues. F-T meat ducks have higher fatty acid synthesis-related gene expression [27], indicating their increased ability for fatty acid synthesis. Therefore, F-T ducks may attain a large amount of fat deposition in adipocytes by synthesizing more fatty acids, leading to fat hypertrophy and proliferation, which results in an increase in adipose tissue.

Lipids are deposited in adipocytes in the form of lipid droplets, which play a positive role in the development of obesity in humans and animals [10,28,29]. The surface of the lipid droplets was covered with a droplet protein, which stabilized it. Recent studies have shown that PLIN1 and PLIN2 are important proteins that regulate lipid droplet lipolysis and lipogenesis [30]. In general, PLIN1 and its cofactor, ABHD5, inhibit lipid lysis by blocking contact between ATGL and lipid droplets [28]. In the stimulated state, PLIN1 is phosphorylated by protein kinase A, which destroys the interaction between ABHD5 and PLIN1, activating ATGL and accelerating lipolysis [31]. Similarly, PLIN2 attenuates lipolysis by reducing ATGL contact with lipid droplets in the basal state [32], whereas lipase cannot be effectively recruited into lipid droplets by PLIN2 protein binding in the stimulated state [33]. Over the past decade, it has been reported that animals deficient in PLIN1 and PLIN2 exhibit a typical lean phenotype with enhanced lipolytic capacity in the body [34,35,36]. In porcine adipose tissue, PLIN1 has been identified as a novel candidate gene affecting IMF (Intramuscular fat) content, PLIN1 knockdown reduces triglyceride (TG) levels and lipid droplet (LD) size in porcine adipocytes [37]. Deletion of PLIN1 mainly enhances lipolysis of adipocytes by exposing lipids in lipid droplets and upregulating lipase [38], while PLIN2 protects its own house, namely lipid droplets. Downregulation of PLIN2 stimulates TG catabolism through autophagy [33], which may provide an explanation for our study, and the lipid metabolism types also produce some differences between the two Pekin ducks due to the differential expression of PLIN1 and PLIN2.ANGPTL4 is mainly involved in energy flow and fat deposition by inhibiting LPL activity and stimulating lipolysis for adipocyte storage. Many studies have reported a positive correlation between circulating ANGPTL4 levels and obesity [39,40]. In the present study, compared with L-T ducks, F-T ducks had higher mRNA levels of PLIN1, PLIN2, and ANGPTL4 in abdominal fat and subcutaneous fat, and showed an increased expression with age. Thus, it has been suggested that F-T ducks and L-T ducks may have different lipid metabolism abilities because of their differences in PLIN1, PLIN2, and ANGPTL4 expression, among which F-T ducks have a higher ability to stabilize lipid droplets, protecting them from lipolysis by ATGL. These results are consistent with the findings of previous studies [34,36]. 

## 5. Conclusions

The present study explored the differences in lipid metabolism between the two strains of Pekin ducks. The data showed that compared to L-T ducks, F-T Pekin ducks could attain more abdominal fat and subcutaneous fat deposition during the growth stage due to higher dietary intake, stronger lipid transport ability, and lipid droplet metabolism regulation. The ABHD5 expression pattern could be associated with the increase in hepatic lipid deposition in L-T ducks, and the gene expression patterns of PLIN1, PLIN2, and ANGPTL4 may explain adipocyte hypertrophy in F-T ducks.

## Figures and Tables

**Figure 1 animals-12-02268-f001:**
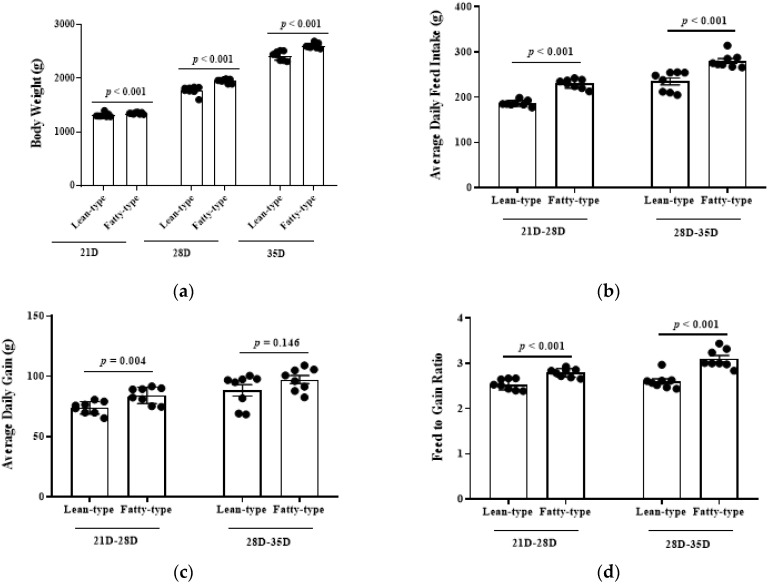
Comparison of body weight (**a**) average daily feed intake (**b**), average daily gain (**c**), and feed-to-gain ratio (**d**) between two strains of meat duck. Values were compared for statistical analysis on three measuring days (21 D, 28 D, and 35 D). Data are given as means and standard error of the mean (*n* = 8). *p*-values are shown after comparison of two strains.

**Figure 2 animals-12-02268-f002:**
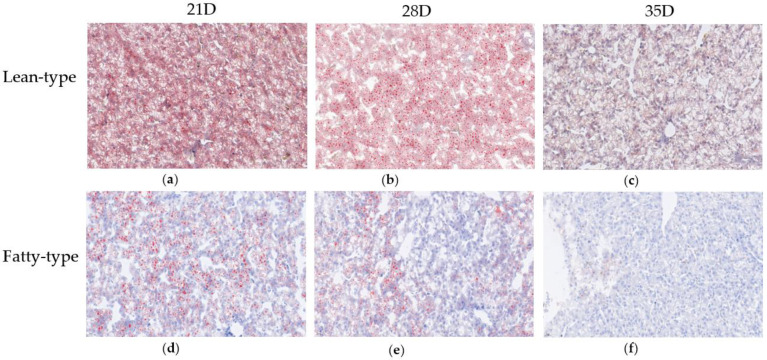
Oil red O staining of liver sections showing the effects of different strains on the hepatic lipid accumulation in the liver of meat duck. Lean-type Pekin ducks after 21 Days of growth (**a**); Fatty-type Pekin ducks after 21 Days of growth (**b**); Lean-type Pekin ducks after 28 Days of growth (**c**); Fatty-type Pekin ducks after 28 Days of growth (**d**); Lean-type Pekin ducks after 35 Days of growth (**e**); Fatty-type Pekin ducks after 35 Days of growth (**f**). The red spots in the pictures are stained lipid droplets. Scale bar: 20 μm.

**Figure 3 animals-12-02268-f003:**
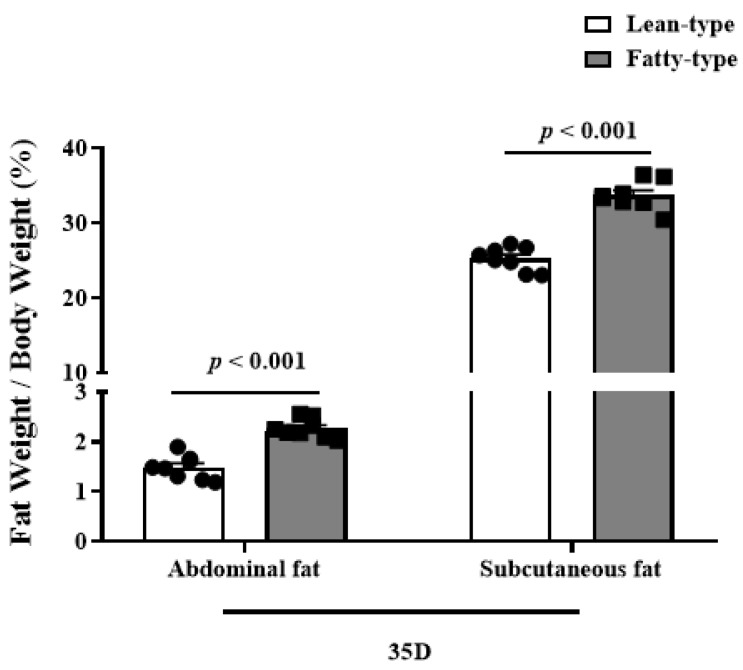
Comparison of abdominal fat and subcutaneous fat contents at 35 D between two strains of meat duck. Data are given as means and standard error of the mean (*n* = 8). *p*-values are shown after comparison of two strains.

**Figure 4 animals-12-02268-f004:**
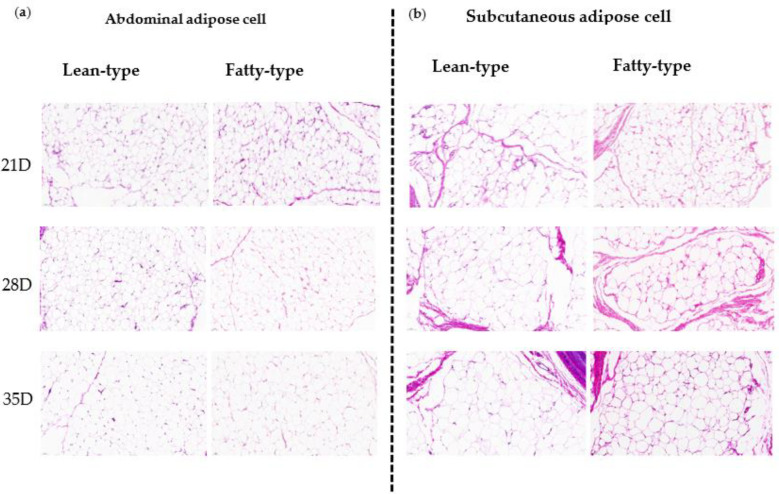
Representative H&E staining pictures of the abdominal adipose cell (**a**) and subcutaneous adipose cell (**b**) of 21 D, 28 D, and 35 D old ducks. The scale bar is 50 μm. Abbreviations: H&E, hematoxylin, and eosin.

**Figure 5 animals-12-02268-f005:**
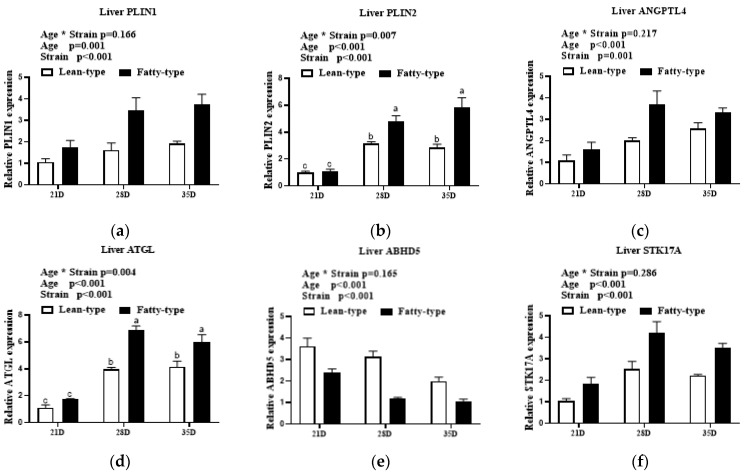
Comparison of expression levels of lipid droplet anabolism genes in liver (**a**–**f**) at 21 D, 28 D, and 35 D between two strains of meat ducks. Data are given as means and standard error of the mean (*n* = 6). Bars with different letters are significantly different (*p* < 0.05). * In the figure represents the interaction between the two factors. Abbreviations: *PLIN1*, perilipin 1; *PLIN2*, perilipin 2; *ANGPTL4*, angiopoietin-like protein 4; *ATGL*, adipose triglyceride lipase; *ABHD5*, abhydrolase domain containing 5; *STK17A*; serine/threonine kinase 17a.

**Figure 6 animals-12-02268-f006:**
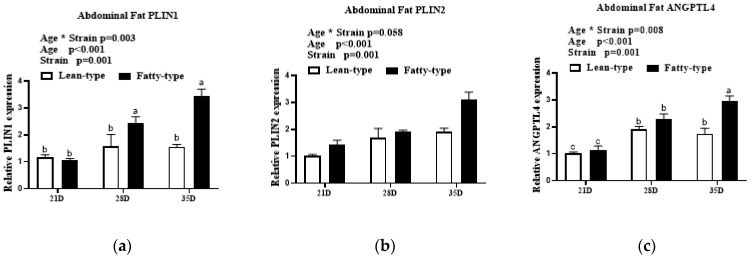
Comparison of expression levels of lipid droplet anabolism genes in abdominal fat (**a**–**f**) at 21 D, 28 D, and 35 D between two strains of meat ducks. Data are given as means and standard error of the mean (*n* = 6). Bars with different letters are significantly different (*p* < 0.05). * In the figure represents the interaction between the two factors. Abbreviations: *PLIN1*, perilipin 1; *PLIN2*, perilipin 2; *ANGPTL4*, angiopoietin-like protein 4; *ATGL*, adipose triglyceride lipase; *ABHD5*, abhydrolase domain containing 5; *STK17A*; serine/threonine kinase 17a.

**Figure 7 animals-12-02268-f007:**
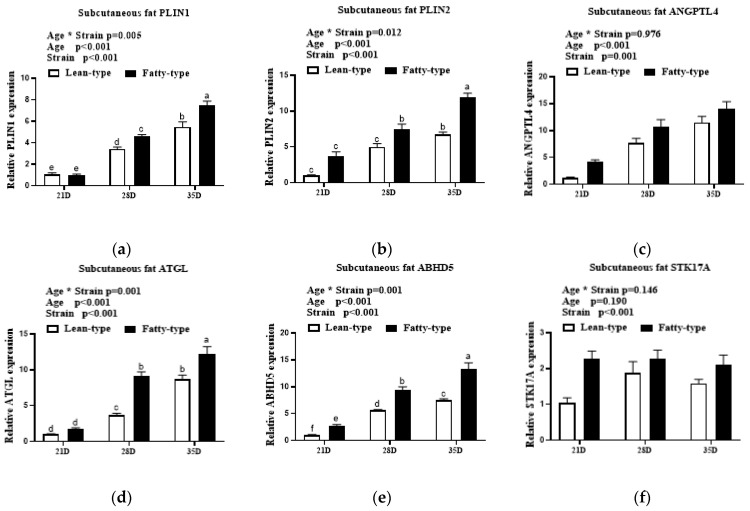
Comparison of expression levels of lipid droplet anabolism genes in subcutaneous fat (**a**–**f**) at 21 D, 28 D, and 35 D between two strains of meat ducks. Data are given as means and standard error of the mean (*n* = 6). Bars with different letters are significantly different (*p* < 0.05). * In the figure represents the interaction between the two factors. Abbreviations: *PLIN1*, perilipin 1; *PLIN2*, perilipin 2; *ANGPTL4*, angiopoietin-like protein 4; *ATGL*, adipose triglyceride lipase; *ABHD5*, abhydrolase domain containing 5; *STK17A*; serine/threonine kinase 17a.

**Table 1 animals-12-02268-t001:** Product guarantee nutritional composition in duckling compound feed (1–35 d).

Item ^1^	Content
Product guarantee value composition (1–21 D), %	
Crude protein	≥19.0
Crude ash	≤8.0
Crude fiber	≤6.0
Calcium	0.5–1.5
Phosphorus	≥0.3
Sodium chloride	0.2–0.8
Methionine	≥0.3
Moisture	≤14.0
Product guarantee value composition (22–35 D), %	
Crude protein	≥16.0
Crude ash	≤9.0
Crude fiber	≤6.0
Calcium	0.9–1.5
Phosphorus	≥0.3
Sodium chloride	0.2–0.8
Methionine	≥0.2
Moisture	≤14.0

^1^ The main raw materials used were corn, soybean meal, corn gluten meal, stone powder, calcium bicarbonate, manganese sulfate, vitamin A, and vitamin B2.

**Table 2 animals-12-02268-t002:** Primer sequence used in this study.

Gene	Sequence (5′→3′)	Product Size (bp)	GenBank Accession
*GAPDH*	F: AGATGCTGGTGCTGAATACG R: CGGAGATGATGACACGCTTA	104	XM_0050106
*PLIN1*	F: GGTATCGGCAGCAGTCTTA R: TTCACAGAGGCGAGTAACTT	200	NM_00131042.1
*PLIN2*	F: CACCACACCGTTAATCTGATCG R: AGTTCTTGACTCTATGTGCTC	171	NM_001310418
*ATGL*	F: TGATGTTATTTACATAGCAATGTC R: TATTAGAAGATATATTTCTGCCAA	157	EU747707
*ABHD5*	F: CCACTTCGACGCTGATGCTC R: ATAAGGTGTTTGACCCTCGAT	168	XM_038174904
*ANGPTL4*	F: CCTGATGGATGCCCAGAACTCCC R: AGACTGCGTTTTGTTGTCCTT	157	XM_038169204
*STK17A*	F: ATTAAACAAGATTTCAAGTGGCT R: TCACTGAAACACCTGCTATGTC	159	XM_013095211

**Table 3 animals-12-02268-t003:** Comparison of serum biochemical parameters between Lean-type ducks and Fatty-type ducks at 35 D.

Item ^1^	Lean-Type	Fatty-Type	*p*-Value
GLU/(mmol/L)	9.12 ± 0.31	9.04 ± 0.22	0.846
TG/(mmol/L)	0.39 ± 0.02	0.39 ± 0.02	0.953
HDL-C/(mmol/L)	2.91 ± 0.11 ^b^	3.30 ± 0.12 ^a^	0.024
LDL-C/(mmol/L)	1.32 ± 0.07 ^b^	1.91 ± 0.14 ^a^	0.001
CHOL/(mmol/L)	5.05 ± 0.21 ^b^	5.90 ± 0.26 ^a^	0.015
UA/(μmol/L)	206.45 ± 9.42	210.30 ± 13.11	0.813
TBIL/(umol/L)	2.03 ± 0.13 ^a^	1.41 ± 0.12 ^b^	0.001
DBIL/(umol/L)	4.72 ± 0.28	4.41 ± 0.18	0.364
TP/(g/L)	44.27 ± 1.18	47.77 ± 2.31	0.186
ALB/(g/L)	17.38 ± 0.38 ^b^	19.24 ± 0.54 ^a^	0.007
LDH/(U/L)	533.65 ± 33.37	582.40 ± 62.00	0.493
CK/(U/L)	1212.15 ± 120.68 ^a^	820.65 ± 39.86 ^b^	0.004
HBDH/(U/L)	681.25 ± 31.01 ^b^	926.45 ± 100.64 ^a^	0.025
ALP/(U/L)	528.30 ± 36.66	448.63 ± 18.17	0.059
ALT/(U/L)	39.96 ± 1.92	40.70 ± 2.51	0.816
AST/(U/L)	41.18 ± 3.45	40.46 ± 5.09	0.906
AST/ALT	1.03 ± 0.07	1.01 ± 0.11	0.858

^1^ Data are presented as the mean and standard error of the mean (*n* = 8). ^a–b^ Means within a row lacking a common superscript differ significantly (*p* < 0.05). Abbreviations: GLU, glucose; TG, triglyceride; HDL-C, high-density lipoprotein cholesterol; LDL-C, low-density lipoprotein; CHOL, cholesterol; UA, uric acid; TBIL, total bilirubin; DBIL, direct bilirubin; TP, total protein; ALB, albumin; LDH, lactate dehydrogenase; CK, creatine kinase; HBDH, hydroxybutyrate dehydrogenase; ALP, alkaline phosphatase; ALT, alanine transaminase; AST, aspartate aminotransferase.

**Table 4 animals-12-02268-t004:** Comparison of adipocytes between Lean-type ducks and Fatty-type ducks at different growth ages.

Growth Ages ^1^	Strain	Abdominal Fat			Subcutaneous Fat		
		Diameter/(μm)	Area/(μm^2^)	Density/(per, mm^2^)	Diameter/(μm)	Area/(μm^2^)	Density/(per, mm^2^)
21 D	Lean-type	59.70	2511.49	392.47	77.00 ^e^	4592.73 ^e^	230.26 ^a^
	Fatty-type	61.37	2844.33	375.84	74.50 ^e^	4328.61 ^e^	236.37 ^a^
28 D	Lean-type	61.60	2913.60	350.93	81.18 ^d^	5056.56 ^d^	218.77 ^a^
	Fatty-type	65.07	3249.38	335.98	86.70 ^c^	5850.40 ^c^	183.42 ^b^
35 D	Lean-type	67.97	3639.73	287.29	91.69 ^b^	6661.35 ^b^	165.11 ^b^
	Fatty-type	71.20	4022.22	252.15	96.28 ^a^	7007.48 ^a^	143.01 ^c^
Pooled SE line		0.69	68.69	8.79	0.85	116.47	7.17
	21 D	60.53 ^c^	2677.91 ^c^	384.15 ^a^	75.75	4460.67	233.31
	28 D	63.33 ^b^	3081.49 ^b^	343.45 ^b^	83.94	5453.48	201.10
	35 D	69.58 ^a^	3830.97 ^a^	269.72 ^c^	93.99	6834.41	154.07
Pooled SE		0.49	48.57	6.21	0.60	82.36	5.07
Strain	Lean-type	63.09 ^b^	3021.61 ^b^	343.56 ^a^	83.29	5436.89	204.72
	Fatty-type	65.88 ^a^	3371.98 ^a^	321.32 ^b^	85.83	5728.83	187.60
Pooled SE		0.40	39.66	5.07	0.49	67.25	4.14
*p*-value (2-way ANOVA)	Growth ages	<0.001	<0.001	<0.001	<0.001	<0.001	<0.001
	Strain	<0.001	<0.001	0.003	0.01	0.002	0.006
	Growth ages * Strain	0.727	0.777	0.450	0.001	<0.001	0.019

^1^ Data represent the means of six replicate individuals (*n* = 6). ^a–e^ Means within a row lacking a common superscript differ significantly (*p* < 0.05). * In the figure represents the interaction between the two factors.

## Data Availability

Data presented in this study are available upon request from the corresponding author.

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
