# Peer review of "Effects of Fatty-Type and Lean-Type on Growth Performance and Lipid Droplet Metabolism in Pekin Ducks"

_animals, 2022, doi:10.3390/ani12172268_

Round 1

Reviewer 1 Report

Title: I recommend rewording the title from a grammatical point of view.

line 20: “fatty and lean Pekin ducks”, this formulation requires clarification. Furthermore, I recommend that the concepts of type and breed be treated correctly. In this manuscript, different metabolic types of the same breed were compared (look at it in line 12).

Keywords: The term "strain" should be supplemented with another word to make sense. The introduction of the word strain repeatedly underlines the consistent use of breed, type, and other groups of individuals within a breed that carry specific characters.

Table 3: for the sake of consistency, in the case of letters written in the superscript, I recommend matching the alphabetical order and the ascending order of the values.

Table 4: check the spelling of the word "anove". Here, I recommend entering the averages with a total of 4 characters. Here, I recommend writing the letters written in the superscript after the average values, because they refer to them (the authors did not deal with analysis of standard error/standard deviation/variance).

line 227-228: check this sentence “However, the interaction between age and strain affected PLIN1, PLIN2, ATGL, and ABHD5 but had no influence on the gene expression of ANGPTL4 and STK17A” in light of results presented in Figure 6.

line 266: put “17” into bracket.

line 305: delete first dot.

It is mentioned in the Introduction that the LT type has a higher proportion of breast meat. Since this is the most significant part of the meat, it can be assumed that LT individuals are heavier at slaughter than FT individuals. Contrary to this, the own study found greater weight gain in FT individuals. For the sake of comprehensibility, it is also necessary to provide information about the tissue composition of the body in the manuscript.

The authors explain why they chose the age of 21-28-35 days. At this age, the ducks are already past usual pre-rearing, but still far from the end of fattening.

The authors do not mention the sex of the test birds or their gender distribution. What do they think about the significant relationship between sex and fat metabolism?

Author Response

Response to Reviewer 1 Comments

Point 1: Title: I recommend rewording the title from a grammatical point of view.

Response 1: Thanks for your kindly comment. We have modified the title.

Point 2: line 20: “fatty and lean Pekin ducks”, this formulation requires clarification. Furthermore, I recommend that the concepts of type and breed be treated correctly. In this manuscript, different metabolic types of the same breed were compared (look at it in line 12).

Response 2: Thanks for your kindly comment. We have modified "fatty and lean Pekin ducks" and modified in "Simple Summary".

Point 3: Keywords: The term "strain" should be supplemented with another word to make sense. The introduction of the word strain repeatedly underlines the consistent use of breed, type, and other groups of individuals within a breed that carry specific characters.

Response 3: Thanks for your kindly comment. We have modified "strain" to "Fatty-type" and

"Lean-type".

Point 4: Table 3: for the sake of consistency, in the case of letters written in the superscript, I recommend matching the alphabetical order and the ascending order of the values.

Response 4: Thanks for your kindly comment. We have matched the alphabetical order and the ascending order of the values in Table 3.

Point 5: Table 4: check the spelling of the word "anove". Here, I recommend entering the averages with a total of 4 characters. Here, I recommend writing the letters written in the superscript after the average values, because they refer to them (the authors did not deal with analysis of standard error/standard deviation/variance).

Response 5: Thanks for your kindly comment. We checked and changed the errors in Table 4 and re-visualised the data.

Point 6: line 227-228: check this sentence “However, the interaction between age and strain affected PLIN1, PLIN2, ATGL, and ABHD5 but had no influence on the gene expression of ANGPTL4 and STK17A” in light of results presented in Figure 6.

Response 6: Thanks for your kindly comment.“However, the interaction between age and strain affected PLIN1, PLIN2, ATGL, and ABHD5 but had no influence on the gene expression of ANGPTL4 and STK17A” are based on Figure 7.

Point 7: line 266: put “17” into bracket.

Response 7: Thanks for your kindly comment. We have revised the errors in the manuscript.

Point 8: line 305: delete first dot.

Response 8: Thanks for your kindly comment. We have revised the errors in the manuscript.

Point 9: It is mentioned in the Introduction that the LT type has a higher proportion of breast meat. Since this is the most significant part of the meat, it can be assumed that LT individuals are heavier at slaughter than FT individuals. Contrary to this, the own study found greater weight gain in FT individuals. For the sake of comprehensibility, it is also necessary to provide information about the tissue composition of the body in the manuscript.

Response 9: Thanks for your kindly comment. We have already provided information on body tissue composition in the introduction.

Point 10: The authors explain why they chose the age of 21-28-35 days. At this age, the ducks are already past usual pre-rearing, but still far from the end of fattening.

Response 10: Thanks for your kindly comment. We found that there was no difference in body size and body composition between the two strains Pekin ducks before 21 days through previous studies, so we chose to observe from 21 days. In some parts of China, Pekin ducks is ready to be marketed at 35 days, we therefore used 35 days as a time node to investigate the effect of strain on lipid metabolism.

Point 11: The authors do not mention the sex of the test birds or their gender distribution. What do they think about the significant relationship between sex and fat metabolism?

Response 11: Thank you for your kindly comment. We mentioned in the abstract the use of males for testing, so it was not stated in the text, and it has now been stated in the text that sex effects on lipid metabolism were avoided by using male ducks.

Reviewer 2 Report

Obs. In Material and Methods the authors mention 224 Pekin ducklings (including 140 FT ducks and 140 LT ducks). 224 is the total number of animals used in the trial? 140 FT plus 140 LT = 280 ducklings. Please clarify and correct this information.

Obs. Table 1. Is nutritional composition, not the value composition.

Obs.  The blood samples collected by wing vein puncture with heparinized syringes equipped with stainless steel needles is according to the animals care standards? Is this procedure unharmful for the animals?

Obs. Row 103 the tissue fixative, what contains?

Obs. Rows 99 – 104. This entire part is confusing and contradictory. For example, the authors mentioned that one bird from each cage was sacrificed. That means 16 birds (8 LT and 8LF) at 21, 28 and 36 days which is equal with 48 ducks / entire experimental period. This are the birds used for abdominal fat and sebum. After, the authors mention that 2 birds /cage were slaughtered to collect the liver, abdominal fat, and sebum, which were fixed in 4 % tissue fixative. Other parts were stored in liquid nitrogen. What parts? The sampling collection is very bad explained. Please re-wrote this part and explain in detail every step.

Obs. Row 105 explain in words first time TG.

Obs. Growth performances. Please add a comment regarding the results of the Feed to Gain parameter.

Obs. Table 3. Please correct the items in the table. FT instead of Title 3.

Obs. Figure 2 should be presented different. For example, make groups of 2 at 21 days, 2 at 28 days and 2 at 35 days, for a better visualization, (as in figure 4) and name them accordingly.

Obs. Why in figure 3 is presented only the comparison of abdominal fat and sebum contents at 35 days? Why the 21 and 28 is missing?

Obs. Enlarge table 4 on the entire page for a better visualization of the data. In this form is too crowded with data. Further, I will suggest to the add the SEM value for each parameter instead of using the std for each one of them. Abbreviations should be moved under the table.

Obs. The discussions section is not enough. I will suggest to explore a little bit more. 

Author Response

Response to Reviewer 2 Comments

Point 1: In Material and Methods the authors mention 224 Pekin ducklings (including 140 FT ducks and 140 LT ducks). 224 is the total number of animals used in the trial? 140 FT plus 140 LT = 280 ducklings. Please clarify and correct this information.

Response 1: Thanks for your kindly comment. We thoroughly checked the materials and methods and modified this error in the manuscript.

Point 2: Table 1. Is nutritional composition, not the value composition.

Response 2: Thanks for your kindly comment. We have modified value composition to nutritional composition.

Point 3: The blood samples collected by wing vein puncture with heparinized syringes equipped with stainless steel needles is according to the animals care standards? Is this procedure unharmful for the animals?

Response 3: Thanks for your kindly comment. Blood sampling by wing vein puncture can reduce the time of blood collection under the premise of obtaining the target blood volume, and reduce the pain and tension time of animals, which is in line with the 3R principle of experimental animals. In addition, the wounds at the wing veins of poultry can be eliminated in a short period of time and are basically not harmful to animals.In the whole process, while completing blood collection, we minimize animal pain, tissue damage and pressure, and strictly abide by animal welfare principles.

Point 4: Row 103 the tissue fixative, what contains?

Response 4: Thanks for your kindly comment. The tissue fixative was 4% paraformaldehyde buffer solution with neutral pH.

Point 5: Rows 99 – 104. This entire part is confusing and contradictory. For example, the authors mentioned that one bird from each cage was sacrificed. That means 16 birds (8 LT and 8LF) at 21, 28 and 36 days which is equal with 48 ducks / entire experimental period. This are the birds used for abdominal fat and sebum. After, the authors mention that 2 birds /cage were slaughtered to collect the liver, abdominal fat, and sebum, which were fixed in 4 % tissue fixative. Other parts were stored in liquid nitrogen. What parts? The sampling collection is very bad explained. Please re-wrote this part and explain in detail every step.

Response 5: Thanks for your kindly comment. We have thoroughly examined this part, rewrote it, and explained every step.

Point 6: Row 105 explain in words first time TG.

Response 6: Thanks for your kindly comment. We have added TG explanation to the manuscript.

Point 7: Growth performances. Please add a comment regarding the results of the Feed to Gain parameter.

Response 7: Thanks for your kindly comment. We have added a comment regarding the results of the Feed to Gain parameter in Growth performances.

Point 8: Table 3. Please correct the items in the table. FT instead of Title 3.

Response 8: Thanks for your kindly comment. We have revised the errors in the manuscript.

Point 9: Figure 2 should be presented different. For example, make groups of 2 at 21 days, 2 at 28 days and 2 at 35 days, for a better visualization, (as in figure 4) and name them accordingly.

Response 9: Thanks for your kindly comment. We have modified figure 2, for a better visualization.

Point 10: Why in figure 3 is presented only the comparison of abdominal fat and sebum contents at 35 days? Why the 21 and 28 is missing?

Response 10: Thanks for your kindly comment. Pekin ducks had less abdominal fat and subcutaneous fat deposition at 21 days and 28 days, were difficult to collect, and operational errors easily caused data to be unavailable, so 35 days abdominal fat and subcutaneous fat percentage were used to represent lipid deposition throughout the growth period.

Point 11: Enlarge table 4 on the entire page for a better visualization of the data. In this form is too crowded with data. Further, I will suggest to the add the SEM value for each parameter instead of using the std for each one of them. Abbreviations should be moved under the table.

Response 11: Thanks for your kindly comment. We have modified table 4, for a better visualization.

Point 12: The discussions section is not enough. I will suggest to explore a little bit more.

Response 11: Thanks for your kindly comment. We have explored the discussion further.